# Coupled binding mechanism of three sodium ions and aspartate in the glutamate transporter homologue Glt_{Tk}

Albert Guskov[1,*], Sonja Jensen[1,*], Ignacio Faustino[1], Siewert J. Marrink[1,2] & Dirk Jan Slotboom[1,2]

Glutamate transporters catalyse the thermodynamically unfavourable transport of anionic amino acids across the cell membrane by coupling it to the downhill transport of cations. This coupling mechanism is still poorly understood, in part because the available crystal structures of these transporters are of relatively low resolution. Here we solve crystal structures of the archaeal transporter Glt_{Tk} in the presence and absence of aspartate and use molecular dynamics simulations and binding assays to show how strict coupling between the binding of three sodium ions and aspartate takes place.

[1] University of Groningen, Groningen Biomolecular Sciences and Biotechnology Institute, Nijenborgh 4, 9747 AG Groningen, The Netherlands. [2] University of Groningen, Zernike Institute for Advanced Materials, Nijenborgh 4, 9747 AG Groningen, The Netherlands. * These authors contributed equally to this work. Correspondence and requests for materials should be addressed to D.J.S. (email: d.j.slotboom@rug.nl).

Glutamate transporters are found in Bacteria, Archaea and Eukarya. In vertebrates they guard the levels of the neurotransmitter glutamate in the synaptic cleft by pumping it (back) into neuronal and glial cells (reviewed in refs 1,2). The free energy stored in cation gradients across the membrane is utilized to accumulate the amino acid inside the cells. The eukaryotic proteins couple glutamate uptake to co-transport of three sodium ions and a proton, and counter-transport of a potassium ion[3–5].

Crystal structures are available for two similar archaeal homologues of the mammalian transporters (Glt_Ph and Glt_Tk, which share 77% sequence identity, Fig. 1a)[6–10]. The archaeal transporters also couple the cellular uptake of aspartate to the co-transport of sodium ions, but they do not use protons or potassium ions for co- or counter-transport[11]. A stoichiometry of three sodium ions per aspartate has been determined for Glt_Ph (ref. 12). The crystal structures have provided a structural framework to explore the mechanism of transport, which has sparked a large amount of biochemical, electrophysiological, mutagenesis and simulation studies on various members of the family. However, the resolution of the structural data has been too low to reveal the binding sites of the sodium ions, and therefore the coupling mechanism has remained poorly understood. Replacement of sodium by thallium ions allowed the use of the anomalous signal from the heavy atom to identify the potential locations of two of the three sodium binding sites in the low resolution structures[9]. Simulation studies indicated possible locations of the third sodium binding site[13–15], but no unequivocal solution was found, leaving the coupling mechanism unexplained.

Here, we present crystal structures, molecular dynamics simulations and substrate binding assays of Glt_Tk to localize the sodium binding sites, and provide insight in the mechanism of coupling between sodium and aspartate binding.

## Results

**Overall structural characteristics of Glt_Tk.** We crystallized the archaeal glutamate transporter homologue from *Thermococcus kodakarensis* (Glt_Tk) in the presence of aspartate and sodium ions, solved its structure at 2.8 Å resolution (Glt_Tk^sub), and located three sodium sites. Isothermal titration calorimetry (ITC) experiments on aspartate binding in the presence of eight different sodium concentrations showed that Glt_Tk indeed binds three sodium ions per aspartate molecule, the same number as Glt_Ph (Fig. 2). In parallel we obtained improved crystals of the substrate-free form of Glt_Tk (ref. 8) that yielded a 2.7 Å resolution structure (Glt_Tk^apo) (see Table 1 for data and refinement statistics). The two structures, each with better resolution and data quality than any previously determined one, provide a structural model for coupling between substrate and cation transport.

Both structures have a very similar trimeric organization with the transport domains in an outward-occluded position (Supplementary Fig. 1). There was well-defined electron density for all loops, as well as for the N-termini, which were not ordered in the Glt_Ph crystals. Notably, we could model the entire loop between transmembrane helices 3 and 4, which is important for transport[16,17] (Fig. 1b). This loop intimately wraps around the outer face of the transport domain, and appears to prevent excessive movements of helical hairpin 2 (HP2) away from the binding pocket, while allowing small-scale movements of HP2, which are required to provide access to the substrate and sodium binding sites (see below).

**Sodium and aspartate binding sites.** For all three sodium ions, as well as the aspartate molecule we observed well defined electron density in the Glt_Tk^sub structure (Fig. 3a–c). The positions of sodium ions in the two sites Na⁺1 and Na⁺2 (numbering of the sodium ion sites as in ref. 9) are in good agreement with the previously reported crystal structures[9], showing that thallium was indeed a good mimic for sodium in these sites (Supplementary

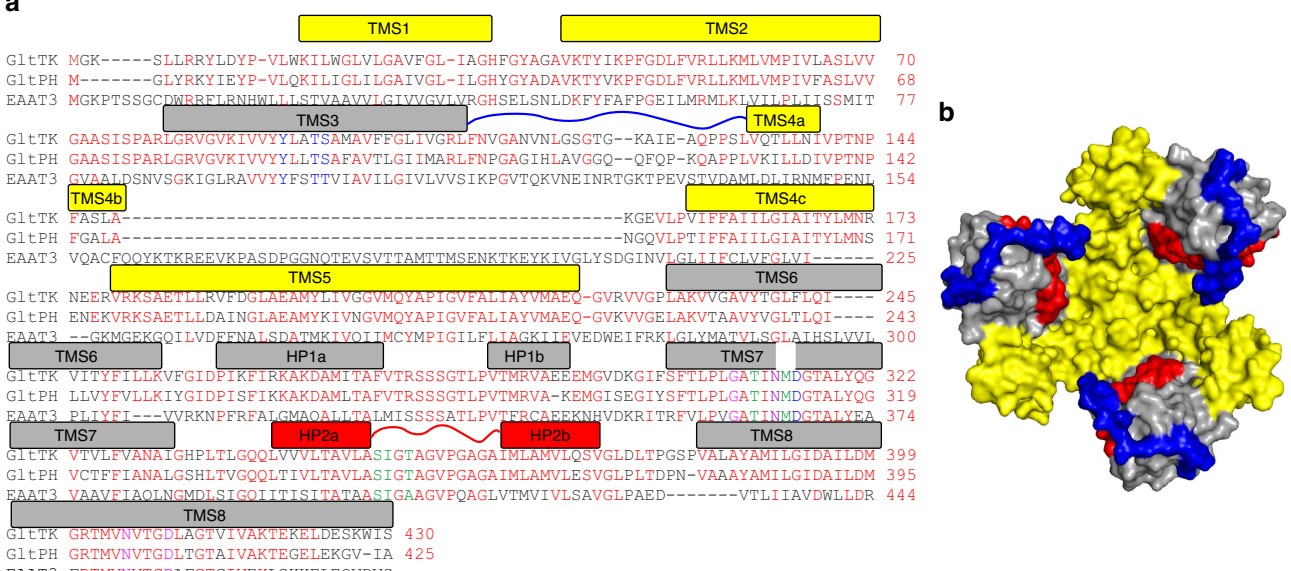

**Figure 1 | Overview of the structure of Glt_Tk.** (**a**) Sequence alignment of the aspartate transporters Glt_Tk form *Thermococcus kodakarensis* and Glt_Ph from *Pyrochoccus horikoschii* and the glutamate transporter EAAT3 from *Rattus norvegicus*. Identical residues between the archaeal proteins colored red, those involved in Na⁺1 binding in magenta, Na⁺2 in green and Na⁺3 in blue respectively. N313 involved in coordination of both Na⁺1 and Na⁺3 is in purple. Bars above the sequences indicate helical segments and are colored in yellow and grey for the trimerization and transport domain, respectively. Loop 3-4 is indicated by the blue line, HP2 in red. (**b**) Crystal structure of the substrate-loaded aspartate transporter Glt_Tk^sub viewed from the extracellular side of the membrane, shown as surface. The trimerization domains in yellow, transport domains in gray, HP2 in red and long loop 3-4 in blue.

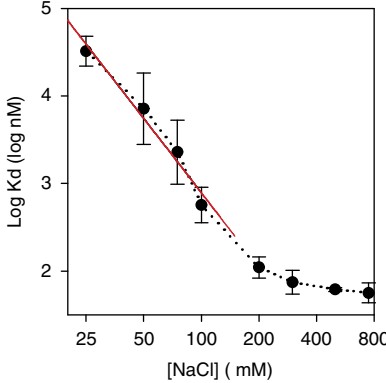

**Figure 2 | Glt$_{Tk}$ binds three sodium ions per aspartate molecule.**
Apparent affinity of Glt$_{Tk}$ for aspartate as a function of Na$^+$ concentration, as determined by isothermal titration calorimetry. In the limit of low sodium concentrations the negative slope of the plot approaches the Na$^+$:aspartate binding stoichiometry[9,10,26]. The slope of the red line is −2.8. Error bars indicate standard deviations from at least 3 independent experiments.

| Table 1 | Data collection and refinement statistics. | |
| --- | --- | --- |
| | Glt$_{Tk}^{apo}$ | Glt$_{Tk}^{sub}$ |
| *Data collection* | | |
| Space group | P3221 | |
| Cell dimensions | | |
| $a$, $b$, $c$ (Å) | 116.01 116.01 308.5 | |
| α, β, γ (°) | 90.00, 90.00, 120.00 | |
| Resolution (Å) | 47.7–2.7(2.78–2.7)* | 48.4–2.8(2.87–2.8)* |
| $R_{meas}$ | 0.06 (0.74) | 0.07 (0.98) |
| $CC_{1/2}$ | 99.6 (36.0) | 99.8 (22.1) |
| $I/\sigma I$ | 11(2.49) | 9.3(1.3) |
| Completeness (%) | 79(17.9) | 97.5 (96.0) |
| Redundancy | 3.4 | 3 |
| | | |
| *Refinement* | | |
| Resolution (Å) | 2.7 | 2.8 |
| No. reflections | 52,820 | 60,568 |
| $R_{work}/R_{free}$ (%) | 19.8/23.7 | 21.3/24.3 |
| No. atoms | | |
| Protein | 9,638 | 9,570 |
| PEG/detergent | 278/40 | 508/99 |
| Ligand/ion | – | 27/9 |
| Water | 60 | – |
| *B-factors* | | |
| Protein | 77.6 | 96.3 |
| PEG/detergent | 114/132 | 118.4/137.7 |
| Ligand/ion | – | 77.6/76.1 |
| Water | 69.9 | – |
| r.m.s. deviations | | |
| Bond lengths (Å) | 0.008 | 0.008 |
| Bond angles (°) | 1.123 | 1.131 |

*Values in parentheses are for the highest-resolution shell.

Fig. 2). Na$^+$1 is coordinated by the β-carboxylate of the conserved D409 residue (D405 in Glt$_{Ph}$, D454 in rat EAAT3, see Fig. 1a for numbering) of Trans Membrane Segment (TMS) 8, and main-chain carbonyls of G309, N313 from the unwound central region of TMS7 and N405 from TMS8 (Fig. 3a and Supplementary Fig. 3). Na$^+$2 is coordinated by main-chain carbonyls (S352, I353 and T355) of HP2 connecting TMS7 and 8, and main-chain carbonyl of T311 (TMS7) (Fig. 3a and Supplementary Fig. 3). The sulfur atom of M314 is within interacting distance (<3 Å) to Na$^+$2, though the nature of this interaction is unclear. The proposal made by Boudker et al.[9] that the Tl$^+$ ion (used as a substitute of Na$^+$) forms favorable interactions with sulfur is unlikely to be the full explanation, because we also see the interaction in the presence of sodium instead of thallium. The unusual coordination of Na$^+$2 by the methionine residue might be necessary for dynamic rearrangements of the binding site during the transport cycle. It was shown previously that the side chain of this methionine points away from the binding site in the substrate-free transporter, indicating that it must be highly mobile[8]. Furthermore, simulations have shown that Na$^+$2 is the last ion to bind during the substrate loading[13,15] and the first one to leave for substrate release, suggesting that the Na$^+$2 binding site might be of low affinity, which is supported by simulations presented below.

**The third sodium binding site.** The third binding site is located between the unwound central region of helix 7 containing the essentially conserved NMDGT motif and TMS3, at a distance of ∼8 Å from Na$^+$1, from which it is shielded by the side chain of N313 (Fig. 3c). Na$^+$3 is coordinated by hydroxyl groups of T94 (T92 in Glt$_{Ph}$, T101 in EAAT3$_{rat}$) and S95 (S93 in Glt$_{Ph}$, T102 in EAAT3$_{rat}$) from TMS3; the carboxamide group of conserved N313 (N310 in Glt$_{Ph}$, N365 in EAAT3$_{rat}$); the side-chain carboxyl of D315 (D312 in Glt$_{Ph}$, D367 in EAAT3$_{rat}$); and the main-chain carbonyl of Y91 (Fig. 3 and Supplementary Fig. 3). The location of the third sodium binding site matches well with the site predicted in one of the published molecular dynamics simulations on Glt$_{Ph}$ (ref. 15), and with mutagenesis studies on mammalian transporters[15,18–21].

Of the three sodium ions, the interaction distances are the shortest for the third ion (∼2.2–2.4 Å compared with ∼2.5–2.7 Å for Na$^+$1 and Na$^+$2), which might indicate tighter binding of Na$^+$3 (Supplementary Table 1). Free energy calculations indeed show that sodium binds more tightly to the

Na$^+$3 than to the two other sites (Supplementary Table 2 and ref. 22). The higher binding affinity may explain why thallium could not replace sodium in the heavy-atom soaking experiment in Glt$_{Ph}$ (ref. 9). Moreover, the observed interaction distances for Na$^+$1 and Na$^+$2 are more similar to those typical for potassium ions, and Tl$^+$ is a very good mimic for K$^+$. It is also possible that steric hindrance prevented the larger Tl$^+$ ion (ionic radius of 1.64 versus 1.16 Å for Na$^+$) to enter the binding area for Na$^+$3.

**Cooperativity between binding of sodium ions and aspartate.** The affinity of Glt$_{Ph}$ and Glt$_{Tk}$ for aspartate is strongly dependent on the sodium concentration[8,9,23]. In the mammalian transporters similar cooperativity between sodium and glutamate has been observed. In addition, sodium binding to Glt$_{Ph}$ in the absence of aspartate is also highly cooperative[23,24]. The structures of liganded Glt$_{Tk}^{sub}$ and a new structure at 2.7 Å resolution of Glt$_{Tk}^{apo}$ provide a structural explanation for these observations. The most prominent difference between the two structures is the conformation of the conserved NMDGT motif in the central unwound region of TMS 7 (Fig. 4a), which is exemplified by the striking relocation of M314 from a position exposed to the lipid bilayer in the apo-state to the position as a ligand of Na$^+$2 in the substrate-bound state (Fig. 4a,c). Sodium binding to N313 plays a crucial role in stabilizing the repositioned NMDGT region. The side-chain of N313 coordinates Na$^+$3, while its main chain carbonyl binds Na$^+$1 (Fig. 4b and Supplementary Fig. 3). Binding of one sodium ion repositions N313, which at the same time optimizes the geometry of the other sodium site, in line with the observed cooperativity in sodium binding.

In the repositioned NMDGT region of Glt$_{Tk}^{sub}$ G316 and T317 are located closer to TMS8 than in Glt$_{Tk}^{apo}$. The position of G316 is stabilized by a hydrogen bond to the side chain carboxamide of

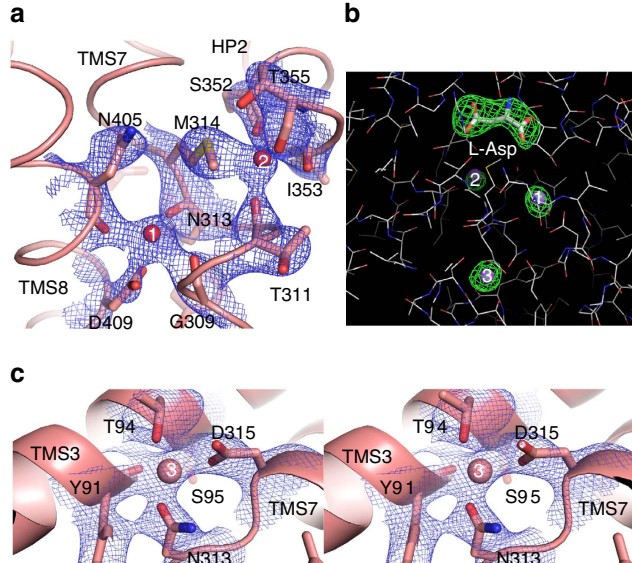

**Figure 3 | Structure of Glt$_{Tk}^{sub}$ sodium binding sites.** (**a**) Sodium binding sites Na$^+$1 and Na$^+$2. The residues forming the sites are shown as sticks, sodium ions as spheres. The 2fo–fc electron density is shown as blue mesh at 3σ. (**b**) Composite omit map (simulated annealing protocol) for the substrates L-Asp (in stick) and sodium ions (purple spheres), contoured at 3.5σ. (**c**) Stereo-view of Na$^+$3 site, depicted as in **a**.

N405 in TMS 8, which in turn binds Na$^+$1 via its backbone carbonyl, again contributing to the strong coupling between the movement of the NMDGT region and sodium binding. The movement places T317 in an optimal position for aspartate binding, which provides a structural explanation for cooperativity between Na$^+$ and aspartate binding. In addition the movement of T317 forces the side chain of R401 to reorient, which creates space for aspartate binding, and places the guanidium group in an optimal orientation to interact with the β-carboxylate of aspartate. In this way sodium binding events in sites Na$^+$1 and Na$^+$3 not only interlink with each other, but also strongly affect the affinity for aspartate. This coupling is further reinforced because binding of sodium to the Na$^+$3 site leads to a movement of TMS3 (which contains three of the Na$^+$3 binding residues) towards TMS7, thereby pushing T317 towards the aspartate binding position (Fig. 4a).

After binding of Na$^+$1, Na$^+$3 and aspartate, Na$^+$2 is likely the last ion to bind[9,13,25]. The binding of Na$^+$2 is coupled to binding of the other two sodium ions and aspartate by the positioning of M314 at the center of the triangle formed by aspartate, Na$^+$1 and Na$^+$2 (Fig. 4b), and by the conformation of the tip of HP2, which interacts both with aspartate and with Na$^+$2. Only after binding of Na$^+$2 can the transport domain reorient to the inward-facing state, because Na$^+$2 binding closes the gate formed by the tip of HP2. MD simulations indeed show that the tip of HP2 is one of the most flexible regions of the protein, and that Na$^+$2 relatively easily leaves this site (Fig. 5 and Supplementary Note 1).

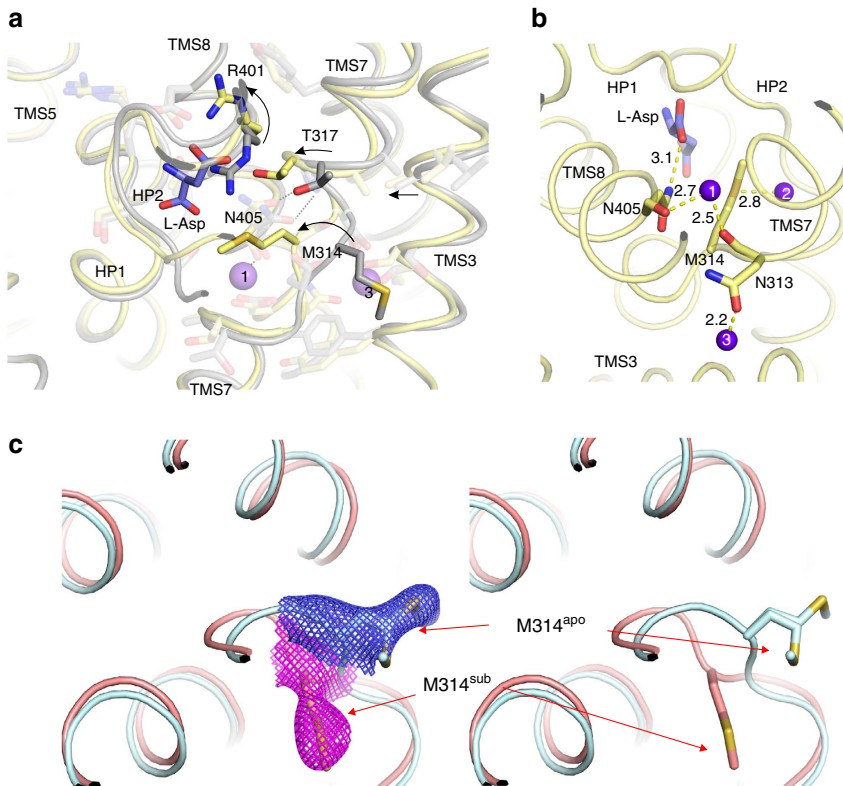

**Figure 4 | Mechanism of sodium-aspartate coupling.** (**a**) Superposition of Glt$_{Tk}^{sub}$ (yellow) and Glt$_{Tk}^{apo}$ (gray). Bound L-Asp shown as blue sticks and sodium ions as purple spheres. Arrows represent the movements during transition from *apo* to the substrate-bound state. Note that parts of the protein which are not directly involved in substrate-binding (for example, TMS2, TMS5) do not undergo any noticeable changes. (**b**) Interaction network between L-Asp (blue sticks) and sodium ions (purple spheres) mediated by N313, M314 and N405 (shown in sticks), distances are given in Å (also see Supplementary Fig. 3). (**c**) M314 in substrate-bound and substrate-free states with 2fo–fc electron density countered at 3σ (left) and without density (right). In the *apo* state two alternative conformations of the side chain are resolved, consistent with the proposed inherent mobility of this residue.

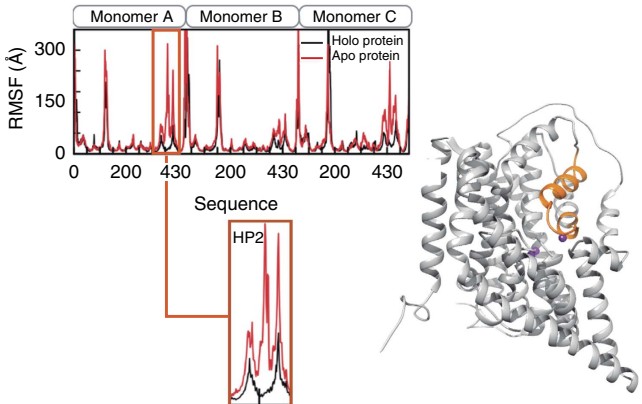

**Figure 5 | HP2 displays enhanced local flexibility.** Comparison of the protein atomic fluctuations averaged per residue in the presence/absence of the ligands in the binding site. Data was averaged from 100 ns MD simulations. The HP2 residues (in orange) involved in the gate opening showed enhanced local flexibility without the ligands (See also Supplementary Fig. 5 for RMSD values of the backbone).

## Discussion

The crystal structures and MD simulations presented here provide insight in the coupled binding of three sodium ions and aspartate. The work is fully consistent with a model for transport in which the cooperative binding of two sodium ions ($Na^{+}1$ and $Na^{+}3$) is followed by aspartate binding and finally binding of $Na^{+}2$. The order in which $Na^{+}1$ and $Na^{+}3$ bind may depend on an obligatory route of sodium ions from the $Na^{+}1$ site to the $Na^{+}3$ site[13]. The postulation of an obligatory route was based on the structure of substrate-bound $Glt_{Ph}$, which showed that the $Na^{+}3$ site could be accessed only via the cavity of the $Na^{+}1$ site. Because the affinity of the $Na^{+}3$ site is higher than that of the $Na^{+}1$ site, the $Na^{+}3$ site would fill up before the $Na^{+}1$ site. Our data are consistent with this binding order, but comparison of the substrate-bound structure with that of the apo-protein (which was not available at the time of the previous simulations) suggests that substantial conformational changes in the NMDGT region must take place upon sodium binding. Neither the $Na^{+}1$ nor the $Na^{+}3$ site is fully formed in the apo state, and it is possible that an alternative access pathway may open upon repositioning of the NMDGT region and movements of the tip of HP2. In that case a random order of the binding of $Na^{+}1$ and $Na^{+}3$ may occur. Further simulations and experiments could clarify the issue. Finally, although the binding of three sodium ions to the sites identified here leads to a stable, crystallizable state, it is possible that also transient binding sites exist, which are briefly occupied while the $Na^{+}$ ions are en-route to their most stable positions[14].

## Methods

**Protein expression and purification.** *E. coli* MC1061 cells containing a pBAD24 derived plasmid for production of C-terminally His$_8$-tagged $Glt_{Tk}$ were cultivated in LB medium (37 °C and 200 r.p.m.), and expression was induced at an $OD_{600}$ of 0.8 with 0.05% L-arabinose for a period of 3 h. Harvesting, membrane vesicle preparation, determination of total protein concentration and solubilisation in n-Dodecyl-β-D-Maltopyranoside was done as described before[12]. To obtain apo-$Glt_{Tk}$ we omitted $Na^{+}$ from all buffers. After solubilisation the solution centrifuged for 30′ at 4 °C, 265,000 g. The supernatant was incubated for 1 h on a rotating platform at 4 °C with Ni-Sepharose (GE Healthcare, 0.5 ml bed volume) pre-equilibrated with 50 mM Tris HCl pH 8, 300 mM KCl, 0.15% n-Decyl-β-D-Maltopyranoside (DM), 15 mM imidazole pH 8. The slurry was poured into a Poly-Prep chromatography column (BioRad), and washed with 10 ml of 50 mM Tris HCl pH 8, 300 mM KCl, 0.15% DM, 60 mM Imidazole pH 8. $Glt_{Tk}$ was eluted with 50 mM Tris HCl pH 8, 300 mM KCl, 0.15% DM, 500 mM Imidazole pH 8, and further purified by size exclusion chromatography (SEC) on a Superdex 200 10/300 GL (GE Healthcare) column in equilibrated with buffer containing 10 mM Hepes KOH pH 8, 100 mM KCl, 0.15% DM. After purification the protein concentration was determined by UV absorption (NanoDrop), using the molecular weight and extinction coefficient calculated by the Expasy Protparam tool (web.expasy.org/protparam/). The protein was concentrated to $\sim 7$ mg ml$^{-1}$ using a spin concentrator (Vivaspin 2, 30,000 MWCO, PES membrane, Vivaproducts). For purification of the substrate-loaded protein solubilization and purification buffers contained 300 mM $Na^{+}$ instead of $K^{+}$, and were supplemented with 10 μM L-aspartate.

**Isothermal titration calorimetry (ITC).** ITC experiments were performed at a constant temperature of 25 °C using an ITC200 calorimeter (MicroCal). L-aspartate (dissolved in buffer containing 10 mM Hepes KOH pH 8, 100 mM KCl, 0.15% DM and indicated sodium concentrations) was titrated into a thermally equilibrated ITC cell filled with 250 μl of 3–20 μM $Glt_{Tk}$ (concentration 13–15 μM), dissolved the same buffer. Data were analysed using the ORIGIN-based software provided by MicroCal.

**Crystallization, data collection and structure determination.** Crystals of $Glt_{Tk}^{apo}$ and $Glt_{Tk}^{sub}$ were obtained with the vapour diffusion technique (hanging drop) with the following conditions: 25% glycerol/ PEG 4000, 100 mM Tris/bicine, pH 8.5, 60 mM CaCl$_2$/MgCl$_2$ and 0.75–1% n-octyl-β-D-glucopyranoside (OG) and 20% PEG 400, 3% Xylitol, 50 mM MgCl$_2$, 150 mM NaCl, 100 mM Glycine pH 9 and 25% glycerol/ PEG 4000, 100 mM Tris/bicine, pH 8.5, 60 mM CaCl$_2$/MgCl$_2$, respectively. Crystals were flash-frozen in liquid nitrogen, cryo-protected with paraton-N and brought to the synchrotron for analysis. Data were collected at beam line X06SA (SLS, Villigen). In addition, we made many attempts to perform anomalous diffraction experiments via crystal soaking or co-crystallization with thallium salts but they did not yield any useful results.

Crystals of $Glt_{Tk}^{apo}$ and $Glt_{Tk}^{sub}$ diffracted up to 2.5 and 2.6 Å resolution; for $Glt_{Tk}^{apo}$ it was not possible to collect fully complete dataset (overall completeness 79%, see Table 1) . Data were processed with XDS (ref. 27), and the structures were solved by Molecular Replacement with Phaser[28] using previously published model of $Glt_{Tk}^{apo}$ (PDB ID 4KY0). Manual rebuilding was done with COOT (ref. 29) and refinement with Phenix refine[30]. Refined models were deposited into PDB repository with the following IDs: 5DWY for $Glt_{Tk}^{apo}$ and 5E9S for $Glt_{Tk}^{sub}$ respectively. Images were prepared using Pymol (Schrödinger, LLC) and Ligplot+ (ref. 31).

**Molecular dynamics simulations.** Two systems were simulated to assess the influence of the ligands in the dynamics of the $Glt_{Tk}$: the *holo* system, which contained the three $Na^{+}$ ions and the aspartate in the binding site, and the same system, from which the ions and aspartate had been omitted. Atomistic MD simulations were carried out using the crystallographic model of $Glt_{Tk}$ (PDB code 4KY0) embedded in a lipid membrane with explicit solvent. To this purpose we used the CHARMM Membrane Builder[32] and converted to Lipid14 PDB format[33]. Each of the original crystal structures was embedded in a lipid bilayer composed of 250 1-palmitoyl-2-oleoylphosphatidylcholine (POPC) lipids, solvated in aqueous solution with TIP3P water molecules[34] and 0.15 M NaCl. The simulation systems contained a total of $\sim 170,000$ atoms. After the initial preparation, the systems were minimized and equilibrated as explained elsewhere[33]. After minimization, coordinates of the protein and ligands were fixed (10 kcal mol Å$^{-2}$) to allow for equilibration of the water and lipid densities, while increasing the temperature up to 303 K. In a second equilibration phase, the systems were relaxed without restraints for 20 ns. MD simulations were run with AMBER 14 (ref. 35, http://ambermd.org/) for another 80 ns using the NPT ensemble (1 atm, 303 K) with periodic boundary conditions and the Langevin thermostat.

**Free energy calculations.** We calculated the relative binding free energy of $Na^{+}$ ions at the Na1, Na2 and Na3 binding sites. In the case of Na1, we considered both the case where no other ions were bound, and the case with the Na3 site already filled. For Na2, assumed to be the last site occupied, all other ligands were included in their binding sites (Supplementary Table 2).

The standard binding free energy can be decomposed as a sum of the translocation and translational free energy contributions:

$$\Delta G_{b} = \Delta \Delta G_{int} + \Delta G_{tr} \qquad (1)$$

where the first term represents the contribution due to the difference in interaction energy of the ligand upon translocation from the bulk to the binding site while the second term represents the change in the translational energy upon binding. The latter is associated to the local fluctuations of the unrestrained $Na^{+}$ bound in the binding site[15,22,36], and was computed according to Heinzelmann *et al.*[22].

The first term, the free energy of translocation, is calculated using thermodynamic integration (TI) by transforming the bound $Na^{+}$ into a water molecule while transforming another water molecule into a $Na^{+}$ ion in the bulk. The transformation is done in three steps: first a discharging step, then a change in van der Waals and bonded interactions, and finally a recharging step. For each step, we used 11 values of λ to go from 0 to 1 by increments of 0.1. Both a forward and a backward transformation were performed. At each λ window, the system was first equilibrated for 0.5 ns and sampled for 1 ns. In all cases, the last snapshot of the unrestrained MD equilibration process was used as initial coordinates for the TI calculations. Calculation of the free energy can run into convergence issues unless the ligand is restrained to the protein through anchoring atoms in the binding site. We therefore

used harmonic distance restraints between the sodium ion and the surrounding protein residues in the binding site. We verified that the effect of these conformational restraints was negligible (in the case of the Na1 and Na3 binding sites around 0.003 kcal mol$^{-1}$, and around 0.60 kcal mol$^{-1}$ for the Na2 binding site).

The convergence of the calculations was assessed by monitoring the cumulative averages of the free energy differences (Supplementary Fig. 4). Given the smooth curves obtained, the free energy change associated to the transformation was calculated using the trapezoidal rule. Translocation free energies were calculated by averaging the values of the forward and backward transformations.

**Data availability**. All relevant data are available from the corresponding author upon reasonable request. The coordinates of the refined models and structure factors have been deposited into the PDB repository: 5DWY for Glt$_{Tk}^{apo}$ and 5E9S for Glt$_{Tk}^{sub}$. The Glt$_{Tk}^{apo}$ structure (PDB ID 4KY0) was used as a molecular replacement model and for MD simulations.

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

## Acknowledgements

We thank A.J.M. Driessen for use of his ITC machine. The research leading to these results received funding from the European Community's Seventh Framework Programme (FP7/2007–2013) under BioStruct-X (to D.J.S. and A.G., grant agreement 283570). This work was funded by the Netherlands Organisation for Scientific Research (NWO) (NWO ECHO grant 711.011.001 and NWO Vici grant 865.11.001 to D.J.S.) and the European Research Council (ERC) (ERC Starting Grant 282083 to D.J.S.). The European Synchrotron Radiation Facility (ESRF) and the Swiss Light Source (SLS) are acknowledged for beamline facilities.

## Author contributions

All authors designed experiments and simulations, analysed data and wrote the manuscript, A.G. and S.J. performed experiments, I.F. performed simulations.

## Additional information

**Competing financial interests:** The authors declare no competing financial interests.

