## [Peer Review File · Nature Communications]

Reviewers' comments:

Reviewer #1 (Remarks to the Author):

The experimental data documented in this manuscript provide new insights in the mechanism of sodium-coupled glutamate transport by an archeal homologue of the glutamate transporters from the brain. The latter are key elements in the termination of glutamatergic synaptic transmission and keep the glutamate concentrations in the synaptic cleft below neurotoxic levels. The homologue has been proven to be an excellent model for the brain transporters. While the location of the acidic amino acid substrate in the binding pocket is known, the location of all three sodium sites has not been firmly established due to the limited resolution of the previously described structures. This report describes two new structures of the homologue GltTk with and without substrate, which have a higher resolution. This enables the assignment of the three sodium sites and provides new insights in the transport mechanism. Therefore this is a very important paper. It is well written and the data are conservatively interpreted. I have only a few suggestions which could further enhance the high quality of the manuscript.

Based on indirect evidence, two locations for sodium site 3 have been proposed. One of them corresponds to the site documented here. Nevertheless, it is possible that there multiple sites are involved in the translocation of the sodium ions and therefore it is possible that the alternative site (see Ref. 14) is used as well but not seen in the crystallized conformation. This issue should be discussed.

Please provide a short explanation on why the sites are assigned as sodium rather than water sites.

What is different in the crystallization conditions used here to enable the improved resolution?

Supplemental Figure 6 is very important because it illustrates the coupling mechanism and should be presented as a regular Figure.

Reviewer #2 (Remarks to the Author):

The two high-resolution crystal structures for the apo and ligand-bound GltTk with three Na⁺ and Asp are important contributions to the glutamate transporters that place the ion-substrate coupling mechanism on a much firmer ground. Two significant results are

- i) Use of the Na⁺ ions instead of Tl⁺, as was done in the previous crystal structures, has allowed direct observation of the Na₃ site and lifted any doubts about the effect of the larger Tl⁺ ions on ion binding sites. The Na₁ and Na₂ sites match well with the sites observed in the earlier crystal structures of GltPh. The newly observed Na₃ site matches exactly with the Na₃ site predicted from MD simulations in GltPh and confirmed by mutagenesis experimenst in Ref. 15 (this is not mentioned in the paper but it should be).
- ii) Comparison of the apo and ligand-bound structures provides important clues on the structural changes that occur in the protein upon ligand binding. These are critical information for a quantitative understanding of the ligand binding mechanism.

I believe these experimental findings are sufficient for the acceptance of the paper. The computations performed to support the experimental results are, unfortunately, not at the same standard and detract from the value of the paper. The worst part are the free energy calculations which are far from

being realistic, e.g., the experimental Na₂ affinity is a few kcal/mol, but the calculated value is 20 kcal/mol. According to the MD simulations (supp. Table 2), Na₂ is not bound in chain B and poorly coordinated in chains A and C. This should have alerted the authors that there is something wrong with the free energy calculations. Both Na₂ and Na₃ free energies are about 18 kcal/mol larger than those given in Ref. 22. This problem is presumably due to the use of restraints in free energy calculations, which can be checked by reducing the strength of the restraints.

In their conclusion, the authors assert that the order in which Na₁ and Na₃ bind may be random but do not provide any evidence for that. This should have been backed up, for example, by calculating the binding free energy of Na₁ but for some reason it was not done. I think the authors need to provide some evidence for this assertion, which contradicts the previous studies (e.g., Refs. 13 and 22) where Na₃ was found to bind first, followed by Na₁.

Minor comment:

In supp. Fig.8, please identify the chains A, B, and C according to the color code. Why is there such a big difference between the apo and ligand-bound RMSDs in the red chain while no visible change occurs in the others?

Reviewer #3 (Remarks to the Author):

The manuscript "Mechanism of coupled transport of three sodium ions and aspartate in the glutamate transporter homologue GltTk." describe crystal structures of bacterial Aspartate transporters which catalyze membrane transport of anionic amino acids in a thermodynamically unfavorable direction coupled to downhill transport of cations. The authors claim that the mechanism of cation/aspartate coupling is poorly understood, in part because the available crystal structures are of relatively low resolution. They provide crystal structures with better data quality of the archaeal aspartate transporter GltTk. The authors have identified three sodium ions and aspartate. Overall, the manuscript is well written and the new findings are important.

However, for my feeling the manuscript needs more experimental evidence for the conclusions.

The authors could provide additional experiments (e.g. radioactivity assays) for sodium/aspartate stoichiometry for this specific aspartate transporter (GltTk) in proteoliposomes or nanodiscs. The authors could discuss at more detail the specific binding of sodium ions (why not other cations?) in aspartate transporters.

The authors could provide also anomalous diffraction experiments with thallium TlNO₃ soaked crystals. This was done for GltPh (Verdon et al, Elife 2014) but not for GltTK. The authors should verify and compare the results of both homologues. Both homologues show "only" a sequence identity of 77%.

The data set of GltTkapo (2.7Å resolution) is surely not complete: Completeness 79% (17.9%). The authors should measure additional data to complete the apo-structure, discuss the obvious problems or remove the data from the manuscript.

Minor issues:

- 1.) The authors could provide additional and important data quality indicators in table 1, for instance Rpim and CC1/2 for the specific shells.
- 2.) The authors could provide detailed figures with simulated annealed omitted maps for all three

potential sodium binding sites and for the aspartate. The density in figure 1b and c looks suspiciously. It would be helpful to see the cation binding sites with different contour levels (for SI).

3.) An overall cartoon figure with all binding sites and the aspartate would be very helpful.

4.) For my feeling the authors could remove the term "glutamate transporter homologue" from the title of the manuscript.

5.) Supplementary figures 4 (overloaded, no 3D feeling) and 5 (density cover the amino acid) are not clear enough. In figure S2 I cannot see any difference between both states? Which method was used?

Reviewer #1 (Remarks to the Author):

The experimental data documented in this manuscript provide new insights in the mechanism of sodium-coupled glutamate transport by an archeal homologue of the glutamate transporters from the brain. The latter are key elements in the termination of glutamatergic synaptic transmission and keep the glutamate concentrations in the synaptic cleft below neurotoxic levels. The homologue has been proven to be an excellent model for the brain transporters. While the location of the acidic amino acid substrate in the binding pocket is known, the location of all three sodium sites has not been firmly established due to the limited resolution of the previously described structures. This report describes two new structures of the homologue Glt_{TK} with and without substrate, which have a higher resolution. This enables the assignment of the three sodium sites and provides new insights in the transport mechanism. Therefore this is a very important paper. It is well written and the data are conservatively interpreted. I have only a few suggestions which could further enhance the high quality of the manuscript.

Based on indirect evidence, two locations for sodium site 3 have been proposed. One of them corresponds to the site documented here. Nevertheless, it is possible that there multiple sites are involved in the translocation of the sodium ions and therefore it is possible that the alternative site (see Ref. 14) is used as well but not seen in the crystallized conformation. This issue should be discussed.

We now discuss this notable possibility on page 8/9 (lines 191-194).

Please provide a short explanation on why the sites are assigned as sodium rather than water sites.

There are several reasons for assigning sodium ions to these sites. First, at 2.8Å resolution we are at the borderline case, when water molecules are usually not yet visible, but light alkali metals are. Since we do not observe densities that could come from ordered water molecules anywhere in the structure, it is likely that the clear densities in the three sites are from sodium ions. Second, the geometries of the sites are fully consistent with sodium binding sites, and we added saturating amounts of sodium. Third, we used apo form of Glt_{TK} for the molecular replacement, thus not inheriting any possible bias from substrate-bound Glt_{Ph} structure. Fourth, we performed many rounds of refinement with simulated annealing and auto-build options (in phenix.refine) to eliminate any bias. Finally, as requested by third reviewer we also included in the revised version the figure with omit map for all three sodium ions.

What is different in the crystallization conditions used here to enable the improved resolution?

The diffraction quality (limit) of a crystal (especially of membrane proteins) is unfortunately the combination of many parameters – we observed differences in diffraction depending on the protein expression/purification batch, batches of

chemicals used for crystallization (e.g. PEGs), conditions during transport of the crystals to the synchrotron, etc. Typically we screen between 100 to 1000 crystals, and only find a few of them showing an improved diffraction.

Supplemental Figure 6 is very important because it illustrates the coupling mechanism and should be presented as a regular Figure.

We have moved Supplemental Figure 6 to the main text as panel **b** in Figure 2.

Reviewer #2 (Remarks to the Author):

The two high-resolution crystal structures for the apo and ligand-bound GltTk with three Na⁺ and Asp are important contributions to the glutamate transporters that place the ion-substrate coupling mechanism on a much firmer ground. Two significant results are

i) Use of the Na⁺ ions instead of Tl⁺, as was done in the previous crystal structures, has allowed direct observation of the Na₃ site and lifted any doubts about the effect of the larger Tl⁺ ions on ion binding sites. The Na₁ and Na₂ sites match well with the sites observed in the earlier crystal structures of GltPh. The newly observed Na₃ site matches exactly with the Na₃ site predicted from MD simulations in GltPh and confirmed by mutagenesis experimentst in Ref. 15 (this is not mentioned in the paper but it should be).

We now refer to this extremely important paper properly on page 5 (lines 109-112).

ii) Comparison of the apo and ligand-bound structures provides important clues on the structural changes that occur in the protein upon ligand binding. These are critical information for a quantitative understanding of the ligand binding mechanism.

I believe these experimental findings are sufficient for the acceptance of the paper. The computations performed to support the experimental results are, unfortunately, not at the same standard and detract from the value of the paper. The worst part are the free energy calculations which are far from being realistic, e.g., the experimental Na₂ affinity is a few kcal/mol, but the calculated value is 20 kcal/mol. According to the MD simulations (supp. Table 2), Na₂ is not bound in chain B and poorly coordinated in chains A and C. This should have alerted the authors that there is something wrong with the free energy calculations. Both Na₂ and Na₃ free energies are about 18 kcal/mol larger than those given in Ref. 22. This problem is presumably due to the use of restraints in free energy calculations, which can be checked by reducing the strength of the restraints.

We thank the reviewer for pointing out this inconsistency. We re-calculated the binding free energies and now have obtained values that are more similar to those previously reported, and more realistic (see Table 2 in the Supplementary Material). The erroneous results we presented before are traced back to a mistake in the input topology files, which affected the absolute values, but not the relative binding strengths. We also evaluated the effect of the constraints, but

these are only minor (less than 1 kcal/mol).

In their conclusion, the authors assert that the order in which Na1 and Na3 bind may be random but do not provide any evidence for that. This should have been backed up, for example, by calculating the binding free energy of Na1 but for some reason it was not done. I think the authors need to provide some evidence for this assertion, which contradicts the previous studies (e.g., Refs. 13 and 22) where Na3 was found to bind first, followed by Na1.

We apologize for the confusion caused by not explaining very well what we meant. We now discuss this point in more detail on page 8. We also have calculated the binding free energy for the Na1 site (Supplementary Table 3). The energy is not as low as the binding free energy for site 3, which would be consistent with a binding order in which site 3 is first filled. However, the *route* to site 3 may not necessarily be via site 1, as previously suggested (discussed on page 8)

Minor comment:

In supp. Fig.8, please identify the chains A, B, and C according to the color code. Why is there such a big difference between the apo and ligand-bound RMSDs in the red chain while no visible change occurs in the others?

A legend with the color code for the chains has been included in Suppl. Fig.8 (new Supplementary Figure 9). The difference in RMSD between the apo and the holo forms for the chain B was associated to the terminal regions. The analysis of the RMSD associated to the non-terminal part showed similar values for each chain.

Reviewer #3 (Remarks to the Author):

The manuscript "Mechanism of coupled transport of three sodium ions and aspartate in the glutamate transporter homologue GltTk." describe crystal structures of bacterial Aspartate transporters which catalyze membrane transport of anionic amino acids in a thermodynamically unfavorable direction coupled to downhill transport of cations. The authors claim that the mechanism of cation/aspartate coupling is poorly understood, in part because the available crystal structures are of relatively low resolution. They provide crystal structures with better data quality of the archaeal aspartate transporter GltTk. The authors have identified three sodium ions and aspartate. Overall, the manuscript is well written and the new findings are important.

However, for my feeling the manuscript needs more experimental evidence for the conclusions.

The authors could provide additional experiments (e.g. radioactivity assays) for

sodium/aspartate stoichiometry for this specific aspartate transporter (GltTk) in proteoliposomes or nanodiscs.

In the revised version we have included new experimental data to determine the sodium:aspartate binding stoichiometry to the Glt_{TK} transporter. We determined the K_d for aspartate binding at eight different sodium concentrations using isothermal titration calorimetry. From these data a stoichiometry of 3:1 was deduced (Supplementary Fig. 2). We decided to do these experiments in detergent solution, as the crystal structures are also obtained in detergent solution, and we focussed our discussion entirely on the coupled binding of aspartate and sodium, rather than on transport.

The authors could discuss at more detail the specific binding of sodium ions (why not other cations?) in aspartate transporters.

We discuss the specificity on page 6 (lines 118-127)

The authors could provide also anomalous diffraction experiments with thallium TINO₃ soaked crystals. This was done for GltPh (Verdon et al, Elife 2014) but not for GltTK.

We have spent more than two years trying to do similar thallium soak experiments with Glt_{TK} but never succeeded.

The authors should verify and compare the results of both homologues. Both homologues show "only" a sequence identity of 77%.

The residues involved in aspartate and sodium binding are identical in both proteins. In addition the transport domain regions in the vicinity of the substrates (second "shell") are almost identical, and the deviations in sequence come from differences in the structures that are far away from the binding sites; nonetheless the overall structures are well conserved. Supplemental Fig 1 provides an overview of the conservation, with the binding site residues indicated.

The data set of GltTkapo (2.7Å resolution) is surely not complete: Completeness 79% (17.9%). The authors should measure additional data to complete the apo-structure, discuss the obvious problems or remove the data from the manuscript.

We agree that the dataset is not complete, however we do not think it should be excluded solely on the ground of completeness. We follow the reasoning of modern approaches to treat the collected data to maximize the useful output from it (as described for example in Karplus PA, Diederichs K. Assessing and maximizing data quality in macromolecular crystallography. *Curr Opin Struct Biol* 2015 34:60-8. doi: 10.1016/j.sbi.2015.07.003). In similar fashion in the manuscript mentioned by the reviewer (Verdon et al, Elife 2014) two structures have completeness of 65.2 (6.5) and 67.9 (13.0)

Minor issues:

1.) The authors could provide additional and important data quality indicators in table 1, for instance Rpim and CC1/2 for the specific shells.

We followed the template of the journal, but now we include this information in

the table 1.

2.) The authors could provide detailed figures with simulated annealed omitted maps for all three potential sodium binding sites and for the aspartate. The density in figure 1b and c looks suspiciously. It would be helpful to see the cation binding sites with different contour levels (for SI).

We now provide the examples of omit maps for all the ligands in SI (Supplementary Fig 4)

3.) An overall cartoon figure with all binding sites and the aspartate would be very We agree, and have included an extra Supplementary Figure 6.

4.) For my feeling the authors could remove the term "glutamate transporter homologue" from the title of the manuscript.

We have a slight preference for keeping the phrase, because Glt_{TK} is member of the "glutamate transporter family".

5.) Supplementary figures 4 (overloaded, no 3D feeling) and 5 (density cover the amino acid) are not clear enough. In figure S2 I cannot see any difference between both states? Which method was used?

We have tried to make a Supplementary Figure 4 clearer (new Supplementary Figure 5). The point of Supplementary figure 5 is to show that the density of Met314 (which undergoes a large movement between the apo- and substrate loaded structures) is well defined. We included a side panel for this figure without density to make it clearer (New Supplementary Figure 7). The goal of Supplementary Figure 2 (new Supplementary Figure 3) is to show that globally the two states (apo and substrate-bound) look very similar; all changes occur in the binding site, which we discuss in the other figures. We added additional description to make it clear in the figure legend. For superposition the SSM protocol was used (Krissinel E, Henrick K (2004). Acta Crystallogr. D60, 2256-226).

Reviewers' comments:

Reviewer #1 (Remarks to the Author):

My only concern is that in the legend to Supplementary Table 3 the four parameters of the binding free energies are not clearly defined or are missing (BWD).

Reviewer #2 (Remarks to the Author):

I am satisfied with most of the responses and the corresponding revision made in the manuscript. In particular, the recalculated free energies have more reasonable values and are also consistent with previous results. There is still some residual problem in the Na2 and Na3 binding free energy calculations (suppl. table 3), where the hysteresis between the forward and backward values are much larger than 1 kcal/mol. This could be due to equilibration problems or use of large spacing in lambda values near 0 and 1. Because this is a side issue, not really related to the main focus of the manuscript, the authors may not want to dwell on it. In that case, a comment alerting the reader to the possible causes of hysteresis would be useful, but I leave the decision to the authors.

Reviewer #3 (Remarks to the Author):

The results reported in the manuscript "Mechanism of coupled binding of three sodium ions and aspartate in the glutamate transporter homologue GlTtk" by Guskov et al. should be interesting to the readership of Nature Communications and the manuscript is well written. In some parts, the authors have satisfactorily addressed my comments.

However, based on the residue-property plots, the quality of both crystal structures for the GlTtk appears to be relatively low. There are 90 (7.0 % for the residues in 5DWY.pdb) and 99 (7.7 % for the residues in 5E9S.pdb) RSRZ residue outliers (real space R-value Z-score). This is an unusual high number of residues which may did not fit to the experimental data (the experimental electron density). Several of the RSRZ values are greater than 4 and I believe the authors could not see any experimental electron density for side chains for such cases. My suggestion for the authors is to cut out the side chains or the complete residue from the PDB-coordinates and re-calculate the structures to reduce significantly the RSRZ residue outliers. The authors should include a sentence that describe and list all missing positions in the methods part of the manuscript. Moreover, all ligands which have a very high LLDF (with a yellow-marked Local Ligand Density Fit) should be revised.

Comments to individual positions in the text:

a) The authors should include a sentence in the methods part in which is clearly stated that they use more or less "incomplete" data for GlTtkapo and that both data sets have a "low" twinning fraction.

b) The authors could include a sentence that all attempts failed to get anomalous diffraction data to verify the potential sodium binding sites (e.g. thallium soaking) which is a key experiment.

c) The authors should provide a new SI-Figure 4 with a "simulated-annealed omitted" map (not an "omitted" map) for all three potential sodium binding sites and for the aspartate (and denote the contour level for any specific map in the figure).

Point-to-point response

We thank all three reviewers once again for the time spent on reviewing our manuscript.

Reviewer #1 (Remarks to the Author):

My only concern is that in the legend to Supplementary Table 3 the four parameters of the binding free energies are not clearly defined or are missing (BWD).

We have modified the legend to the table.

Reviewer #2 (Remarks to the Author):

I am satisfied with most of the responses and the corresponding revision made in the manuscript. In particular, the recalculated free energies have more reasonable values and are also consistent with previous results. There is still some residual problem in the Na₂ and Na₃ binding free energy calculations (suppl. table 3), where the hysteresis between the forward and backward values are much larger than 1 kcal/mol. This could be due to equilibration problems or use of large spacing in lambda values near 0 and 1. Because this is a side issue, not really related to the main focus of the manuscript, the authors may not want to dwell on it. In that case, a comment alerting the reader to the possible causes of hysteresis would be useful, but I leave the decision to the authors.

We have added a comment on the issue in the legend of Supplementary Table 3

Reviewer #3 (Remarks to the Author):

The results reported in the manuscript "Mechanism of coupled binding of three sodium ions and aspartate in the glutamate transporter homologue GltTk" by Guskov et al. should be interesting to the readership of Nature Communications and the manuscript is well written. In some parts, the authors have satisfactorily addressed my comments.

However, based on the residue-property plots, the quality of both crystal structures for the GltTk appears to be relatively low. There are 90 (7.0 % for the residues in 5DWY.pdb) and 99 (7.7 % for the residues in 5E9S.pdb) RSRZ residue outliers (real space R-value Z-score). This is an unusual high number of residues which may did not fit to the experimental data (the experimental electron density). Several of the RSRZ values are greater than 4 and I believe the authors could not see any experimental electron density for side chains for such cases. My suggestion for the authors is to cut out the side chains or the complete

residue from the PDB-coordinates and re-calculate the structures to reduce significantly the RSRZ residue outliers. The authors should include a sentence that describe and list all missing positions in the methods part of the manuscript. Moreover, all ligands which have a very high LLDF (with a yellow-marked Local Ligand Density Fit) should be revised.

We welcome that this reviewer has high quality standards, but we disagree with the suggestion that the quality of the structures is low. On the contrary, the quality is very good as seen from the validation reports. For the substrate-loaded structure and apo structure, respectively: Rfree: 0.24 and 0.236, Clashcore: 8 and 5, Ramachandran outliers: 0 and 0, Side chain outliers: 2.2% and 1.7%.

The reviewer takes into account only the relatively high RSRZ values of 7.7% and 7%. In general, RSRZ is not a perfect measuring tool for the model's quality and rather serves as an indicator of POSSIBLE problems with a model. The best current view on this issue is summarized by one of the most renowned experts in the field Prof Dr Randy Read in his post to ccp4 bulletin board <https://www.jiscmail.ac.uk/cgi-bin/webadmin?A3=ind1602&L=CCP4BB&E=quoted-printable&P=4999730&B=-Apple-Mail%3D7DFF65B2-77B2-48F1-A8E9-4963CF97617B&T=text%2Fhtml;%20charset=windows-1252>

In addition, we noticed that the values for RSRZ depend on the software used for refinement with Refmac giving lower values for RSRZ than Phenix. Such differences have also been noted by others.

Regarding the LLDF values, they have been regularly checked during the refinement, and we noticed that the higher values correspond well with the flexible ligands, such as PEG and detergent molecules, which in general are less ordered than protein residues or specific ligands (Na⁺ and Asp), causing the higher values. Omitting the PEG and detergent molecules from the model leads to strong positive peaks (blobs) in fo-fc maps, and therefore we decided to keep these ligands in. Most importantly, none of the specific ligands (Na⁺ and Asp) have the high LLDF, so we believe it is not a significant issue.

For all side-chains included in our model there is experimental electron density, albeit indeed those residues that show high RSRZ are less defined. As an example we show a screenshot of the density for Trp16 in chain B, which has RSRZ score of 10. It is not perfect density, but it is still there.

Comments to individual positions in the text:

a) The authors should include a sentence in the methods part in which is clearly stated that they use more or less "incomplete" data for GltTkapo and that both data sets have a "low" twinning fraction.

We included the sentence about completeness of data sets (lines 311-313), the twinning fraction (0.02) is negligible, and data quality checking programs report that no twinning is suspected, so we do not think it is necessary to mention.

b) The authors could include a sentence that all attempts failed to get anomalous diffraction data to verify the potential sodium binding sites (e.g. thallium soaking) which is a key experiment.

We included such a sentence (lines 308-310).

c) The authors should provide a new SI-Figure 4 with a "simulated-annealed omitted" map (not an "omitted" map) for all three potential sodium binding sites and for the aspartate (and denote the contour level for any specific map in the figure).

We included the revised composite-omit map with simulated annealing figure (calculated with phenix) into supplement. The peaks in this map for ligands are well-defined.

REVIEWERS' COMMENTS:

Reviewer #3 (Remarks to the Author):

The authors have satisfactorily addressed my comments. I would recommend a publication.